# Recent Advances in the Application of Antibacterial Complexes Using Essential Oils

**DOI:** 10.3390/molecules25071752

**Published:** 2020-04-10

**Authors:** Tae Jin Cho, Sun Min Park, Hary Yu, Go Hun Seo, Hye Won Kim, Sun Ae Kim, Min Suk Rhee

**Affiliations:** 1Department of Food and Biotechnology, College of Science and Technology, Korea University, 2511, Sejong-ro, Sejong 30019, Korea; microcho@korea.ac.kr; 2Department of Biotechnology, College of Life Sciences and Biotechnology, Korea University, 145, Anam-ro, Seongbuk-gu, Seoul 02841, Korea; ash101@korea.ac.kr (S.M.P.); bluet219@korea.ac.kr (H.Y.); gomi960625@korea.ac.kr (G.H.S.); kgpdnjs@korea.ac.kr (H.W.K.); 3Department of Food Science and Engineering, Ewha Womans University, Seoul 03760, Korea; sunaekim@ewha.ac.kr

**Keywords:** natural antimicrobial agent, antimicrobial effect, anti-infectious effect, combined treatment, antibacterial complex, antibacterial mode-of-action, disinfectant, emulsion, antibacterial synergism, antibacterial antagonism

## Abstract

Although antibacterial spectrum of essential oils (EOs) has been analyzed along with consumers’ needs on natural biocides, singular treatments generally require high concentration of EOs and long-term exposures to eliminate target bacteria. To overcome these limitations, antibacterial complex has been developed and this review analyzed previous reports regarding the combined antibacterial effects of EOs. Since unexpectable combined effects (synergism or antagonism) can be derived from the treatment of antibacterial complex, synergistic and antagonistic combinations have been identified to improve the treatment efficiency and to avoid the overestimation of bactericidal efficacy, respectively. Although antibacterial mechanism of EOs is not yet clearly revealed, mode of action regarding synergistic effects especially for the elimination of pathogens by using low quantity of EOs with short-term exposure was reported. Whereas comprehensive analysis on previous literatures for EO-based disinfectant products implies that the composition of constituents in antibacterial complexes is variable and thus analyzing the impact of constituting substances (e.g., surfactant, emulsifier) on antibacterial effects is further needed. This review provides practical information regarding advances in the EO-based combined treatment technologies and highlights the importance of following researches on the interaction of constituents in antibacterial complex to clarify the mechanisms of antibacterial synergism and/or antagonism.

## 1. Introduction

Essential oils (EOs) and EO components are mainly secondary metabolites that are volatile aromatic products extracted from plants (e.g., herbs, spices) [1,2]. EOs have been reported to have numerous bioactivities including antioxidation effects [3] and anti-inflammatory effects [4]. This has led to their use in embalming, in pharmaceutical formulas, or as food additives. In particular, EOs have been regarded as considerably effective antibacterials and anti-infectious agents from natural sources in various fields including the food, medical, pharmaceutical, public health, and environmental fields [5].

In addition, EOs have been shown to have antibacterial [6,7], antifungal [8,9], antiviral [10], antimycotic [11], antiparasitic [12], and insecticidal activities [13]. Along with the drastically increased interests from consumers in the use of natural agents for the development of antimicrobial products (e.g., disinfectants, preservatives, and food additives), natural compounds including EOs are preferred and the role of EOs as alternatives to synthetic chemical agents has been emphasized [14,15,16]. The majority of EOs are listed as generally recognized as safe (GRAS) substances [17].

Previous studies have focused on the discovery of novel EOs with high antibacterial effects [18,19,20]. Since the antimicrobial efficacy (i.e., microbiocidal/microbiostatic activity and spectrum) of EOs varies based on the treatment conditions (e.g., the extraction methods of the EO from natural sources, temperature, treatment concentration, and surrounding compounds), studies on the development of decontamination technologies and/or the optimization of treatment conditions are ongoing [21,22,23]. However, the application of EOs can be limited by the following factors: (1) The higher costs compared with using synthetic agents, (2) the need for high concentrations to achieve bacteriostatic effect (the inhibition of bacterial growth without killing cells) or bactericidal effects (the destruction of bacterial cells) [24], and (3) the adverse effects after the EO treatment (e.g., changes in the physicochemical and sensory characteristics of the subject of application) [18,25]. Thus, the major hurdle in broadening the applicability of EOs is the development of technologies to improve their antibacterial effects.

The development and subsequent application of antibacterial complexes is one of the most representative strategies for improving the decontamination efficacy of EOs [26]. Examining the efficacy of EO-based antibacterial complexes is a prerequisite for the evaluation of the efficiency of their combinations because combined treatments can either increase or decrease their actual effects. Comprehensive analysis of the accumulated findings that highlights these unexpected shifts in antibacterial effects is needed to apply combined treatment technologies in practice and prevent the overestimation of the antibacterial effects by avoiding treatment conditions that result in antagonism.

Previous studies reported that the formulation of EOs as antibacterial complexes showed unexpected improvements in effectiveness and/or efficiency (i.e., antibacterial synergism), and the development of technologies specific to the formulation of antibacterial complexes has been regarded as a novel direction for advancing this field. Although literature reviews regarding the antibacterial effects of EOs are available [18,27,28], reviews focusing on synergism validated by quantitative microbiological analysis from short-term treatment are rarely reported [23,29]. This review covers the current issues regarding antibacterial complexes of EOs, which can be divided into the following sections: (1) Background information on the decontamination effects when EOs are used as constituents of antibacterial complexes; (2) combined treatments of EO-based antibacterial complexes with additive, synergistic, or antagonistic effects; (3) bactericidal mechanisms of combined treatments using EOs inducing synergistic effects; and (4) practical applications of EO-based antibacterial complexes.

## 2. EOs as Antimicrobial Agents

EOs are among the most representative natural antimicrobial agents, and they are widely used as decontaminants in addition to being used as additives and preservatives. Topics of previous reports regarding EOs as antimicrobial agents are diverse according to the purpose of use (e.g., decontamination, elimination of pathogens, delay of use-by-date, and preservation). EOs extracted from plant parts (e.g., seeds, flowers, buds, herbs, and roots) have been used as antimicrobial agents [30]. Among the numerous kinds of EOs, marked bactericidal activity against pathogens has mainly been shown for crude oils extracted from plants, β-resorcylic acid (RA), carvacrol (CAR), cinnamaldehyde (CA), eugenol (EUG), *trans*-cinnamaldehyde (TC), thymol (TM), and vanillin (VNL) [18,31]. The biocidal properties of EOs have also been reported, especially their broad spectrum activities against various bacterial species (e.g., *Acinetobacter baumanii*, *Aeromonas sobria*, *Bacillus cereus*, *Clostridium perfringens*, *Cronobacter sakazakii*, *Enterococcus faecalis*, *Escherichia coli*, *Klebsiella pneumoniae*, *Listeria innocua*, *Listeria monocytogenes*, *Paenibacillus larvae*, *Proteus* spp., *Pseudomonas aeruginosa*, *Salmonella* Enteritidis, *Salmonella* Typhimurium, *Serratia marcescens*, *Staphylococcus aureus*, and *Vibrio parahaemolyticus*) [18,32,33,34,35,36,37,38,39]. To evaluate the antibacterial performance of EOs, several researchers have primarily investigated the minimum inhibitory concentration (MIC; i.e., the lowest concentration of a chemical that prevents visible growth of bacteria) and/or minimum bactericidal concentration (MBC; i.e., the lowest concentration of an antibacterial agent required to kill bacteria) of the EO against pathogens [32,33,34,35,40,41]. Table 1 summarizes the antimicrobial effects of EOs as a singular treatment (i.e., not in combination with other substances). Since the definitions of MIC and MBC differ between researchers, it might not be possible to directly compare the results from those studies. Moreover, since both the MIC and MBC are determined by exposing the target pathogen to the EO for a sufficient time to ensure bacterial growth in the control group (i.e., without any EO treatment) [42,43], a time-dependent quantitative microbiological analysis to identify an actual active concentration is unavailable.

Evaluations of antimicrobial effects from the perspective of the practical application of EOs have been conducted based on the quantitative analysis of the reduction of a microbial population (i.e., log reduction) after EO treatment (Table 2). However, the limitation of a singular EO treatment in the efficiency has been highlighted, as reported by previous studies, mainly because of: (1) The negligible antibacterial effects (*ca.* <1 log reduction) [25,44,45], (2) the requirement of a high EO concentration to achieve a desirable effect from short-term exposure [46,47], or (3) the requirement of a long-term exposure [48,49,50,51,52,53,54,55,56,57,58,59].

In addition, food matrix is a representative case for emphasizing these limitations in the practical application of EOs because higher amounts of EOs and/or long-term exposure are generally required to achieve sufficient bactericidal effects (Table 3), as found from in vitro experiments conducted by using bacterial suspensions (Table 2) [18,60]. This phenomenon can likely be attributed to the complexity of food matrices and/or the presence of available nutrients that support the recovery of injured bacteria [61]. Previous studies on the occurrence of synergistic effects in EO-based antibacterial complex revealed that the antibacterial effect of singular treatment with an EO at a concentration that achieved synergism in a combined treatment is negligible (*ca.* <1 log reduction) [30,62]. Even previous studies that achieved desirable antibacterial effects (e.g., the delay of food spoilage by reducing natural flora) noted that concerns remain about the adverse changes in the quality of the target foods [63]. While the elevation of treatment temperature (e.g., 50–60 °C) was considered to complement the low antimicrobial effect by the addition [64] or the vaporization of EOs [65]. Since using high amount of EOs may accompany with unpleasant odor and a burden of cost (i.e., high price), novel treatment methods without any quality change are needed and the combined treatment can be a representative countermeasure.

## 3. Antibacterial Complex Using EOs

The development and the application of a technological basis for combined treatments using EOs have focused on the unexpected effects of the formulation of antibacterial complexes. The use of combinations of multiple antimicrobial agents can result in various combined effects according to the composition and concentration of the components [74,75,76] (Figure 1): (1) Synergistic effects: antimicrobial activity of the blend of antimicrobial that is greater than the sum of the effects of the individual components, (2) additive effects: the antimicrobial activity is equal to the sum of the effects of the individual components, (3) antagonistic effects: the antimicrobial activity is less than the sum of the effects of the individual components.

As shown in Table 4, recent studies on antibacterial complexes have mainly reported synergistic or antagonistic effects rather than additive effects. The aims of developing combined treatment technologies inducing antibacterial synergism are mainly broadening the applicability of EOs: 1) The maximization of the antimicrobial effects and 2) the optimization of treatment conditions especially for major factors influencing treatment efficiency (e.g., shortening treatment time, manipulating the composition of antibacterial complex specialized for stress-adapted and/or stress-tolerant pathogens). Since EOs are representative compounds obtained from natural sources (e.g., plants), the combination of natural agents was considered a primary technological hurdle, and antibacterial complexes formulated from several EOs were developed [36,37]. Pei et al. [36] reported the antibacterial synergism between EUG and three other EOs (CA, CAR, TM) against *E. coli*; antibacterial mechanisms of the combinations could be hypothesized that CAR and TM may have disintegrated the outer membrane of the target pathogen [17,32], helping EUG to easily enter the cytoplasm and combine with proteins [77]. It was also suggested that the synergism from EUG + CA was based on their action on diverse proteins or enzymes in bacterial cells.

Formulations consisting of EOs with other natural antimicrobials were also suggested including medium chain fatty acids (MCFAs) [25,62], organic acids (OAs) [37], MCFAs + OAs (caprylic acid + citric acid) [78], citrus fruit extracts [45], nisin [79,80,81,82], and foodstuffs [e.g., sodium chloride (NaCl), soy sauce, and teriyaki sauce] [30,44,83]. Enhancing the antibacterial effects of EOs by inducing synergism was also achieved with chelating agents (e.g., EDTA) [37] and nanomaterials [e.g., biological silver nanoparticles (bio-AgNPs)] [67]. In the case of antagonistic effects, researchers have focused on the occurrence of undesirable decreases in the antibacterial effects of combined treatments to facilitate 1) the prevention of overestimating the EOs’ efficacy, and 2) the establishment of countermeasures against these unexpected combined effects prior to practical application [29,74,76]. Combined effects can vary based on the target bacterial species, highlighting the importance of the evaluation of antibacterial complexes for each target bacterial species; as shown by research on the complex formulated from lauric alginate with EO, which showed synergistic effects against *L. monocytogenes* but antagonistic effects against *E. coli* O157:H7 and *S.* Enteritidis [84].

## 4. Antimicrobial Mechanisms of EO Complex against Pathogens

The modes of action (MOAs) of the antimicrobial effects of EOs have been investigated from the perspective of the interactions between EOs and the target microorganisms. Although the mechanisms are not yet fully understood and remain controversial, key principles have been reported for representative EOs (e.g., CAR, TM, EUG, and TC). These principles include: (1) Disrupting the outer membrane; (2) causing cell lysis or the release of lipopolysaccharides; (3) changing the fatty acid composition of the membrane; (4) dissolving, aligning, or forming channels in the phospholipid bilayer; (5) interfering with or inhibiting glucose uptake; and (6) inhibiting enzyme activity [18,85]. Previous studies regarding the MOA of the antibacterial effects of EOs have suggested that those mechanisms involve changing the characteristics of the membrane [17,31,32,86,87]. The specific membrane changes by EOs have mainly been attributed to the destruction of the membrane based on the damage to the cytoplasmic membrane (e.g., CAR and TM) [17] or the alteration of the membrane fatty acid composition (e.g., EUG and TC) [88].

In the case of MOAs regarding the EO-based antibacterial complexes, most previous relevant studies have focused on the mechanisms of the combined effect derived from the growth inhibition caused by long-term exposure (e.g., using the MIC test, and checkerboard assay) [18,23,26,29,42,89,90]. Since the general aim of combined treatments is to achieve synergistic activity by using the smallest quantity of EOs with short-term exposure [14], the investigation of the MOA of synergistic bactericidal effects validated by quantitative microbiological analysis is regarded as primary information in the research fields of antibacterial complexes. Direct comparison of the characteristics for target bacterial cells subjected to singular treatments that showed negligible bactericidal effects and combined treatment showing dramatic synergism is expected to provide key evidence for the antibacterial synergism.

Since the cell membrane disruption is a major antibacterial mechanism of EOs, researches on the MOA of EO-based complexes have also focused on membrane integrity of bacteria after the singular or combined treatment [44,62]. According to the research by Choi et al. [62], time-dependent changes in membrane integrity analyzed by flow cytometry showed gradual increases in the population of permeabilized cells (i.e., membrane-disrupted cells) for both *C. sakazakii* and *S.* Typhimurium following combined treatment with caprylic acid + VNL (Figure 2). Because flow cytometry can be used to demonstrate the MOA of EOs, comprehensive analysis of the results of flow cytometry in conjunction with TEM was adopted for elucidating the mechanisms of combined effects relative to those of singular treatments; this strategy was used to study *S. aureus* treated with CAR + NaCl [44]. As shown in Figure 3, comprehensive analysis of flow cytometry plots and TEM images indicates that the target cells treated by antibacterial complexes have characteristics distinct from cells treated with the components of these complexes. TEM images following singular treatments with CAR and NaCl showed disrupted membranes with damage to the cytoplasm and a decrease in the density of the cytoplasm, respectively. However, most of those cells maintained their morphological characteristics despite the slight increases in damaged cells observed by flow cytometry, highlighting that membrane damage caused by singular treatments using low quantity of EOs with short-term exposure was reversible and insufficient for affecting viability [9]. The combined treatments showed evidence of irreversible damage or cell death by both analytical methods (flow cytometry and TEM) based on the increases in the population of damaged/dead cells and highly deformed membranes allowing the leakage of cellular material from the cells, respectively.

## 5. Practical Application of EO-Based Antibacterial Complex

The development of disinfectants in the form of emulsions or antibacterial films are representative applications of antibacterial complexes. Commercial disinfectant products are generally formulated with various materials other than EOs, and their antibacterial effects are likely to vary based on the amounts and identities of those components [29,91,92]. However, as shown in Table 5, most previous studies regarding the evaluation of the antimicrobial efficacies of the disinfectants have focused on the direct evaluation of the end-product of the formulation rather than the contribution of each component from antibacterial complex. Since EOs incorporated into antibacterial complexes are typically crude extracts rather than EO components (e.g., CA, CAR, and EUG) [93,94], unfortunately, the comparative analysis of the findings from those studies and the identification of the impact from incorporating each ingredient other than EO (e.g., the bonding agent, surfactant, thickener, emulsion stabilizer, emulsifier, ointment base, preservative, film former, plasticizer, detergent, cation, and organic substances) are unavailable.

However, studies on the differences in the antibacterial activities of EO-based emulsions and free EOs alone have reported that the influences of the composition and/or constituents of the emulsions on the antibacterial characteristics of the EOs varied [100,105,109,111]. The evaluation of the antibacterial efficacies of various formulations of emulsions containing *Ocimum gratissimum* leaf oil revealed the factors that can improve the antibacterial effect, namely, increasing the Ocimum oil content and decreasing the content of the surfactant (Tween 80) [109]. Water-based emulsion systems suppressed the antimicrobial activity of garlic EO, and this negative effect was attributed to the relatively small water-soluble fraction of this EO [100]. In contrast, the antibacterial effect presented by the EOs alone against methicillin-resistant *S. aureus* was increased by emulsification with rhamnolipids [111]. No difference in the antibacterial activities of the free EO (thyme oil) and emulsion (thyme oil emulsions containing soluble soybean polysaccharide as an emulsifier) was reported based on the identical MIC and MBC values against *S.* Enteritidis, *E. coli* O157:H7, and *L. monocytogenes*, which indicated that the emulsifier did not affect the antibacterial properties of EO [105].

The effects of other types of components on the antimicrobial activities of EO have also been reported [95,110]. Patrone et al. [110] observed synergy in antibacterial effects against *P. aeruginosa* when eucalyptus and mint oils were combined with methylparaben used as a preservative. Synergistic antibacterial effects against *S. aureus* induced by combinations of preservatives (propylparaben and imidazolidinyl urea) with mint and oregano EO were also reported [110]. Investigations of the influences of organic matter (sheep blood, horse serum, bovine serum albumin, dry bakers’ yeast, and skim-milk powder), surfactants (Tween 20, Tween 80, alkyl dimethyl betaine, and sodium monododecyl sulphate), and cations (Ca^2+^ and Mg^2+^) on the antimicrobial activity of tea tree oil showed that the incorporation of organic matter and surfactants compromised the antimicrobial efficacy of tea tree oil, although certain variations between organisms were observed [95].

## 6. Conclusions

To overcome the disadvantages of EOs (e.g., weak bactericidal effects, high price, and unpleasant odors), most studies regarding antibacterial complexes aim to achieve synergistic effects through combined treatments. This review provides comprehensive information regarding the findings and implications of using EOs as natural antibacterial agents, especially from the perspective of antibacterial complexes showing synergistic effects. However, most studies evaluating the antibacterial effects of EOs (in singular or combined treatments) and examining their MOAs are based on long-term exposure to pathogens. To encourage the practical application of EO, novel technologies that can eliminate target pathogens with short-term treatment by synergism from a small amount of antibacterial agent and a MOA linked to the combined effects should be developed. Since the formulation of EO-based disinfectant products can determine efficacy, subsequent studies on the interactions of the constituents of antibacterial complexes are expected to reveal key combinations for improving practical effects. Moreover, in-depth analyses on the combined effects of multiple agents in antibacterial complexes and the environmental conditions that can affect their efficiencies enables the identification of the optimum compositions for disinfectant products. This focus review provides practical information for the application of EO-based antibacterial complexes in the fields of food, public health, medical science, and pharmacology.

## Figures and Tables

**Figure 1 molecules-25-01752-f001:**
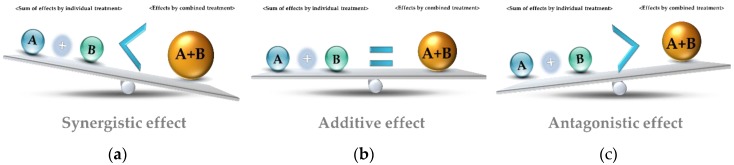
Combined effects of antibacterial complex composed with multiple antimicrobial agents: synergistic effect (**a**); additive effect (**b**); antagonistic effect (**c**).

**Figure 2 molecules-25-01752-f002:**
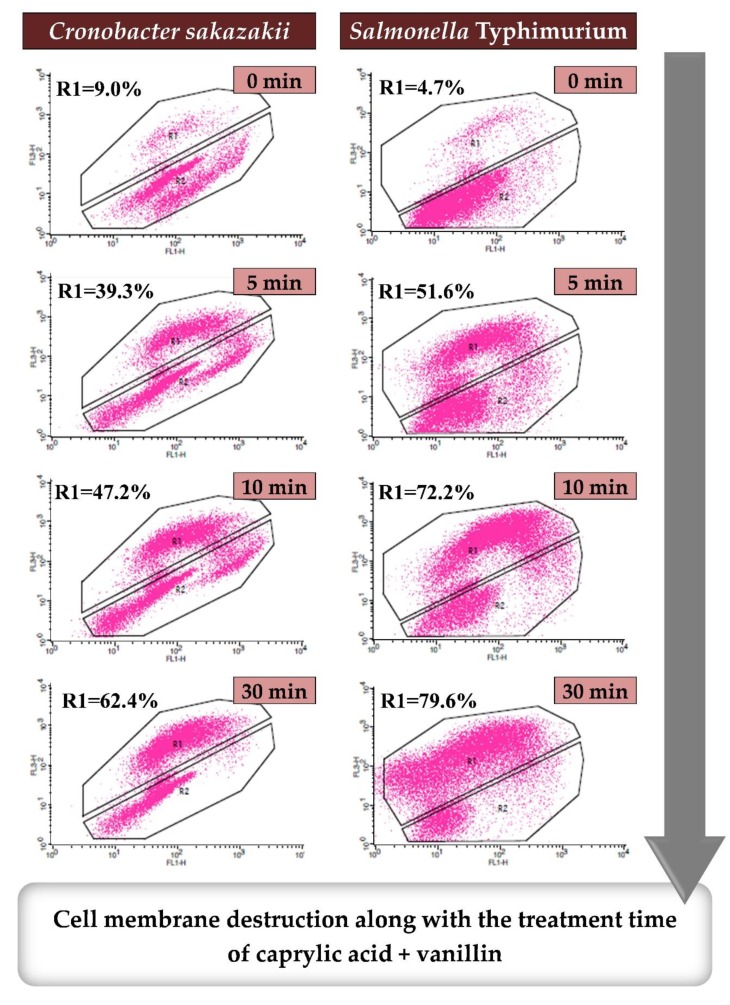
Time-dependent flow cytometry plots for the demonstration of the cell membrane destruction by the EO-based antibacterial complex against *Cronobacter sakazakii* and *Salmonella* Typhimurium treated with caprylic acid + vanillin. This figure was adopted from Choi et al. [62].

**Figure 3 molecules-25-01752-f003:**
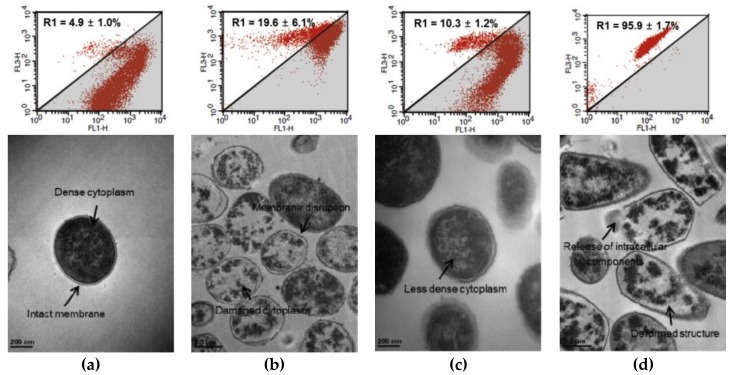
Comprehensive analysis of flow cytometry plots and TEM images for demonstrating the mode of action for the synergistic effects of essential oil-based antibacterial complex against *Staphylococcus aureus* treated with singular treatment (carvacrol or NaCl) and combined treatment (carvacrol + NaCl): control (untreated cells) (**a**), singular treatment of carvacrol (2.0 mM) (**b**), singular treatment of NaCl (15.0%) (**c**), carvacrol (2.0 mM) + NaCl (15.0%) (**d**). This figure was adopted from previous research reported by Kim et al. [44].

**Table 1 molecules-25-01752-t001:** Minimum inhibitory concentration of representative essential oils.

Essential Oils	Target Microorganisms	Antimicrobial Effects (Minimum Inhibitory Concentration; MIC; μL/mL)	References
Carvacrol	*Escherichia coli*	0.225–0.4	[32,36,37]
	*Escherichia coli* O157:H7	3	
	*Salmonella* Typhimurium	0.225–0.25	
	*Listeria monocytogenes*	0.375–5	[33]
	*Staphylococcus aureus*	0.175–0.45	[41]
	*Bacillus cereus*	0.1875–0.9	[38]
Thymol	*Escherichia coli*	0.225–0.4	[32,36]
	*Salmonella* Typhimurium	0.056–0.25	
	*Listeria monocytogenes*	0.45	[33]
	*Staphylococcus aureus*	0.14–0.225	[41]
	*Bacillus cereus*	0.45	[40]
Eugenol	*Escherichia coli*	1.0–1.6	[18,33,36]
	*Escherichia coli* O157:H7	1.7	
	*Salmonella* Typhimurium	0.5	
	*Listeria monocytogenes*	>1.0	[34]
*Trans*-cinnamaldehyde	*Escherichia coli*	0.382–1	[32,36,37,39]
	*Escherichia coli* O157:H7	0.52	
	*Salmonella* Typhimurium	0.382–1	
	*Listeria monocytogenes*	3.82	[34]
Vanillin	*Escherichia coli*	2.183	[35]
	*Listeria innocua*	5.093	

**Table 2 molecules-25-01752-t002:** Antimicrobial efficacy of singular treatment of essential oils in microbial suspension.

Medium	Treatment Conditions	Target Microorganisms	Singular Treatment ^1^	Antibacterial Effects (Log Reduction; Log CFU/g or mL)	References
0.85% saline	37 °C, 10 min	*Escherichia coli* O157:H7	CAR/EUG/RA/TC/TM/VNL 1 mM	negligible (*ca.* <1)	[25]
	22 °C, 5 min	*Escherichia coli* *Staphylococcus aureus*	CAR/TM 2 mM	negligible (*ca.* <1)	[45]
Deionized water	22 °C, 10 min	*Escherichia coli* *Listeria monocytogenes* *Staphylococcus aureus*	TM 2 mM	negligible (*ca.* <1)	[44]
	22 °C, 10 min	*Escherichia coli* *Listeria monocytogenes* *Staphylococcus aureus*	CAR 2 mM	1–2	
Deionized water (with 10 μg/μL Tween 80)	1 min ^2^	*Escherichia coli* O157:H7 *Listeria innocua*	CAR 0.875 μg/mL	>4	[46]
0.1% peptone water	37 °C, 30 min	*Escherichia coli* O157:H7	cinnamon bark oil 0.0625%	>3	[66]
			cinnamon leaf oil 0.0625%	>3	
		*Salmonella* Typhimurium	cinnamon bark oil 0.0625%	4	
			cinnamon leaf oil 0.0625%	>3	
Brain heart infusion broth	4 °C, 8 h	*Escherichia coli*	*Mentha arvensis* L. oil 0.625 μL/mL	>5	[49]
		*Salmonella* Enteritidis	*Mentha piperita* oil 5 μL/mL	>5	
	37 °C, 8 h	*Escherichia coli*	armoise oil 0.10%	>8.0	[50]
			clove oil 0.10%	>7.5	
Butterfield’s phosphate buffer	2 min ^2^	*Escherichia coli* *Salmonella* Typhimurium *Staphylococcus aureus*	orange oil 10%	7	[47]
Fish peptone broth	4 °C, 12 d	*Listeria monocytogenes*	*Bunium persicum* (Black zira) oil 0.20%	3.2	[52]
Luria-Bertani broth	22 °C, 3 h	*Escherichia coli* O157:H7 *Cronobacter sakazakii*	TM 150 μg/mL	1	[59]
			CAR 300 μg/mL	1	
			TC 350 μg/mL	1	
Mueller-Hinton broth	4 °C, 24 h	*Campylobacter jejuni*	rosemary extract 310 μg/mL	7	[51]
	37 °C, 0.17 h	*Escherichia coli* *Staphylococcus aureus*	oregano oil 0.596 μg/mL	5	[67]
Phosphate-buffered saline	37 °C, 72 h	*Salmonella* Typhimurium	bark cinnamon oil 0.5%	>9	[48]
	37 °C, 8 h	*Listeria monocytogenes*	leaf cinnamon oil 0.5%	>9	
	2 min ^2^	*Vibrio parahaemolyticus*	orange oil 10%	7	[47]
Tryptic soy broth	32 °C, 24 h	*Listeria monocytogenes*	bark cinnamon oil 313 ppm	2.0	[49]
			TM 625 ppm	5.3	
	37 °C, 16 h	*Escherichia coli* O157:H7	noni oil 4 μL/mL	>8	[54]
		*Salmonella enterica*	noni oil 4 μL/mL	>8	
	32 °C, 24 h	*Escherichia coli* O157:H7	clove oil 600 μg/mL	>5	[55]
		*Salmonella* Typhimurium	garlic/cinnamon oil 600 μg/mL	3	
		*Listeria monocytogenes*	garlic/clove oil 400 μg/mL	>5	
	37 °C, 24 h	*Escherichia coli*	*Eucalyptus globulus* oil 5 μL/mL	8.7	[56]
		*Salmonella* Enteritidis	*Eucalyptus globulus* oil 7.5 μL/mL	8.1	
		*Bacillus cereus*	*Eucalyptus globulus* oil 5 μL/mL	9.0	
		*Staphylococcus aureus*			
	35 °C, 3 h	*Escherichia coli* *Staphylococcus aureus*	TM 300 ppm	4–5	[57]
		*Listeria monocytogenes*	TM 500 ppm	4–5	
		*Salmonella* Typhimurium			
	37 °C, 24 h	*Escherichia coli*	EUG/VNL 125 μg/mL	7	[58]

^1^ Abbreviation of essential oils: carvacrol (CAR), eugenol (EUG), β-resorcylic acid (RA), *trans*-cinnamaldehyde (TC), thymol (TM), and vanillin (VNL). ^2^ Treatment temperature was not indicated in the previous reports.

**Table 3 molecules-25-01752-t003:** Antimicrobial efficacy of singular treatment of essential oils in foods.

Matrix	Treatment Condition	Target Microorganisms	Singular Treatment ^1^	Antibacterial Effects (Log Reduction; Log CFU/g or mL)	Reference
Soy sauce	22 °C, 10 min	*Escherichia coli* O157:H7 *Salmonella* Typhimurium *Listeria monocytogenes*	CAR/TM 1 mM	negligible (*ca.* <1)	[30]
Infant formula (reconstituted)	45 °C, 30 min	*Cronobacter sakazakii* *Salmonella* Typhimurium	VNL < 30 mM	negligible (*ca.* <1)	[62]
Ground beef	Heat (60 °C, 1 h), vacuum package	*Clostridium perfringens*	CAR/TM/CA/oregano oil 0.1–2.0%	3.2–5.0	[64]
Ground beef	Marination with wine, storage (5 °C, 10 d)	*Salmonella enterica* *Listeria monocytogenes*	oregano oil 0.5%	1.0–3.1	[68]
Catfish fillet	Storage (4 °C, 14 d)	*Listeria monocytogenes*	CAR/thyme oil/oregano oil 1–5%	<4	[69]
Taramosalata	Storage (4, 10 °C, 9 d)	*Salmonella* Enteritidis *Listeria monocytogenes*	mint oil 0.5–2.0%	1.1–1.9	[70]
Mozzarella cheese		*Listeria monocytogenes*	clove oil 0.5–1%	1–3	[71]
Alfalfa seed	60 °C (1, 3, 7 h)	*Salmonella* spp.	TM/CA 200–600 μg/mL of air	>3	[65]
Honeydew	Storage (4 °C, 21 d)	Natural flora	CA 5–15 mM	<5.1	[63]
Lettuce/baby carrot	1–15 min	*Escherichia coli* O157:H7	thyme oil 0.1–10.0 μg/mL	1.5–2.0	[72]
Boiled rice		*Bacillus cereus*	CAR 0.15–0.75 μg/mg	1.0–3.8	[73]

^1^ Abbreviation of essential oils: carvacrol (CAR), thymol (TM), cinnamaldehyde (CA), eugenol (EUG), and vanillin (VNL).

**Table 4 molecules-25-01752-t004:** Efficacy of combined treatment of essential oils as antibacterial complex.

Components of the EO-Based Antibacterial Complex	Treatment Conditions	Target Microorganisms	Combined Treatment ^1^	Antibacterial Effects (Log Reduction; Log CFU/g or mL) [combined effect]	Reference
Combination of EOs	37 °C, 24 h	*Escherichia coli*	CA 100 mg/L + TM 100 mg/L	2.2 [Synergism]	[36,37]
			CA 100 mg/L + CAR 100 mg/L	2.1 [Synergism]	
			TM 100 mg/L + CAR 100 mg/L	2.4 [Synergism]	
	37 °C, 24 h	*Salmonella* Typhimurium	CA 50 mg/L + TM 100 mg/L	0.44 [Synergism]	[37]
			CA 50 mg/L + CAR 100 mg/L	0.42 [Synergism]	
			TM 100 mg/L + CAR 100 mg/L	0.27 [Synergism]	
Medium chain fatty acid	37 °C, 24 h	*Escherichia coli* O157:H7	capric acid 0.4 mM + RA/CAR/EUG/TM/TC 0.4 mM	>7 [Synergism]	[25]
			caprylic acid 1.0 mM + RA/CAR/EUG/TM/TC 1.0 mM		
			lauric acid 0.5 mM + RA/CAR/TM 1.0 mM		
	40 °C, 10 min	*Cronobacter sakazakii*	caprylic acid 20 mM + VNL 30 mM	>7 [Synergism]	[62]
	40 °C, 5 min	*Salmonella* Typhimurium	caprylic acid 20 mM + VNL 30 mM		
Organic acid	37 °C, 24 h	*Salmonella* Typhimurium	lactic acid 0.10% + CAR 200 µL/L	0.37 [Synergism]	[37]
			acetic acid 0.05% + TM 100 mg/L	0.57 [Synergism]	
			acetic acid 0.05% + CAR 100 µL/L	0.15 [Synergism]	
Caprylic acid + citric acid	3 °C, 10 d	*Listeria monocytogenes*	0.5% caprylic acid + 0.1% citric acid + 0.2% oregano oil	<5 [Synergism]	[78]
Citrus fruit extracts	22 °C, 5 min	*Escherichia coli* O157:H7 (Acid-adapted)	calamansi 10% + CAR/TM 2.0 mM	>6.9 [Synergism]	[45]
		*Salmonella* Typhimurium (Acid-adapted)	calamansi/lemon 10% + CAR/TM 2.0 mM		
		*Listeria monocytogenes* (Acid-adapted)	calamansi/lemon/lime 10% + CAR/TM 2.0 mM		
Lauric arginate (LAE)	21 °C, 48 h	*Listeria monocytogenes*	LAE 375 ppm + cinnamon leaf oil/EUG/TM 3,000 ppm	>4 [Synergism]	[84]
		*Escherichia coli* O157:H7	LAE 375 ppm + cinnamon leaf oil/EUG 2,500 ppm	>2 log growth ^2^ [Antagonism]	
			LAE 375 ppm + TM 2,000 ppm	>2 log growth ^2^ [Antagonism]	
		*Salmonella* Enteritidis	LAE 375 ppm + cinnamon leaf oil/EUG 2,500 ppm	>1 log growth ^2^ [Antagonism]	
			LAE 375 ppm + TM 2,000 ppm	>2 log growth ^2^ [Antagonism]	
Nisin	8 °C, 20 min	*Listeria monocytogenes*	nisin 5.3 µg/mL + CAR 1.3 mmol/L	*ca.* 3 log reduction [Synergism]	[79]
	8 °C, 30 min	*Bacillus cereus*	nisin 5.3 µg/mL + CAR 0.7 mmol/L	*ca.* 3 log reduction [Synergism]	
	4 °C, 12 d	*Listeria monocytogenes*	nisin 1000 IU/g + thyme essential oil 0.6%	4.0 log reduction [Synergism]	[80]
	4 °C, 12 d	*Salmonella* Enteritidis	nisin 500 IU/g + oregano essential oil 0.9%	*ca.* 4 log reduction [Synergism]	[82]
	37 °C, 32 h	*Escherichia coli* O157:H7	nisin 500 IU/g + thyme essential oil 0.6%	*ca.* 1 log reduction [Synergism]	[81]
EDTA	37 °C, 24 h	*Salmonella* Typhimurium	EDTA 75 mg/L + TM 100 mg/L	0.7 log reduction [Synergism]	[37]
Sodium chloride	22 °C, 1 min	*Escherichia coli* O157:H7	sodium chloride 5% + CAR 2.0 mM	7 log reduction [Synergism]	[44]
		*Listeria monocytogenes*			
		*Staphylococcus aureus*	sodium chloride 10% + CAR 2.0 mM		
	22 °C, 1 min	*Escherichia coli* O157:H7	sodium chloride 3% + TM 2.0 mM	7 log reduction [Synergism]	
		*Listeria monocytogenes*	sodium chloride 10% + TM 1.0 mM		
		*Staphylococcus aureus*	sodium chloride 15% + TM 1.0 mM		
Soy sauce	4 °C, 5 min	*Escherichia coli* O157:H7	soy sauce + TM 0.5 mM	7 log reduction [Synergism]	[30]
	4 °C, 5 min	*Listeria monocytogenes*	soy sauce + TM 0.5 mM		
	4 °C, 10 min	*Salmonella* Typhimurium	soy sauce + TM 0.5 mM		
Teriyaki sauce	4 °C, 7 d	*Escherichia coli* O157:H7	teriyaki sauce + TM/CAR 0.5%	3.0–3.4 log reduction [Synergism]	[83]
		*Listeria monocytogenes*			
		*Salmonella* Typhimurium			
Biological silver nano particles (bio-AgNPs)	37 °C, 24 h	Methicillin resistant *Staphylococcus aureus*	bio-AgNP 125 µM + *Origanum vulgare* oil 0.298 mg/mL	>5 log reduction [Synergism]	[67]
		*Escherichia coli*	bio-AgNP 31.25 µM + *Origanum vulgare* oil 0.075 mg/mL	>5 log reduction [Synergism]	

^1^ Abbreviation of essential oils: cinnamaldehyde (CA), thymol (TM), carvacrol (CAR), β-resorcylic acid (RA), eugenol (EUG), and *trans*-cinnamaldehyde (TC).

**Table 5 molecules-25-01752-t005:** Major ingredients and antibacterial efficacy of EO-based disinfectant composites.

Species	Product Type	No. of Components Other than EO ^1^	EO with Antibacterial Activity ^2^	Test Method	Reference
*Acinetobacter baumanii*	EO + Interfering substance	1	tea tree oil	Agar diffusion, broth dilution	[95]
*Aeromonas sobria*	EO + Interfering substance	1	tea tree oil	Agar diffusion, broth dilution	[95]
*Bacillus cereus*	Carboxymethyl cellulose film	2	*Zataria multiflora Boiss* oil	Agar diffusion	[96]
*Escherichia coli*	Emulsion	8	*Thymus vulgaris*, *Origanum onites*	Agar diffusion	[97]
	Corn and wheat starch film	7	cinnamon, lavender, lemongrass, lemon oil, peppermint, tea tree	Agar diffusion	[98]
	Wound dressing films	3	lemon oil	Agar diffusion	[99]
	Carboxymethyl cellulose film	2	*Zataria multiflora Boiss* oil	Agar diffusion	[96]
	Water-based emulsion	2	garlic oil	Agar diffusion	[100]
	Cream formulation	7	*Lavandulla officinallis*, *Melaleuca alternifolia*, *Cinnamomum zeylanicum* oils	Time-kill assay	[101]
	Chitosan film	3	*Eucalyptus globulus* oil	Agar diffusion	[102]
	Emulsion	2	lemongrass, majoram, clove, palmarosa, tea tree, rosewood, thyme, sage, geranium, mint	Time-kill assay	[103]
	Gelatin film	2	oregano, lavender oil	Agar diffusion	[104]
	Cellulose film	2	CA, EUG	Vapor diffusion	[93]
*Escherichia coli* O157:H7	Surfactant micelles	1	CAR, EUG	Broth dilution	[94]
	Emulsion	1	thyme oil	Broth dilution	[105]
	Chitosan film	3	oregano oil	Agar diffusion	[106]
*Enterococcus faecalis*	Emulsion	8	*Origanum onites*	Agar diffusion	[97]
	EO + Interfering substance	1	tea tree oil	Agar diffusion, broth dilution	[95]
*Klebsiella pneumoniae*	EO + Interfering substance	1	tea tree oil	Agar diffusion, broth dilution	[95]
*Listeria monocytogenes*	Surfactant micelles	1	CAR, EUG	Broth dilution	[94]
	Water-based emulsion	2	garlic oil	Agar diffusion	[100]
	Edible coating	2	ginger oil	Agar diffusion	[107]
	Emulsion	1	thyme oil	Broth dilution	[105]
	Cellulose film	2	CA, EUG	Vapor diffusion	[93]
	Chitosan film	3	oregano oil	Agar diffusion	[106]
*Paenibacillus larvae*	EO + Emulsifier	1	wild chamomile, Andean thyme oil	Broth dilution	[108]
*Pseudomonas aeruginosa*	Carboxymethyl cellulose film	2	*Zataria multiflora Boiss* oil	Agar diffusion	[96]
	Cream formulation	7	*Lavandulla officinallis*, *Melaleuca alternifolia*, *Cinnamomum zeylanicum* oils	Time-kill assay	[101]
	Chitosan film	3	*Eucalyptus globulus* oil	Agar diffusion	[102]
	EO + Interfering substance	1	tea tree oil	Agar diffusion, broth dilution	[95]
*Proteus* spp.	Topical formulation	1-4	*Ocimum gratissimum* leaf oil	Agar diffusion	[109]
*Staphylococcus aureus*	Emulsion	8	*Thymus vulgaris*, *Origanum onites*	Agar diffusion	[97]
	Topical formulation	1-4	*Ocimum gratissimum* leaf oil	Agar diffusion	[109]
	Wound dressing films	3	lemon oil	Agar diffusion	[99]
	Topical formulation	1-4	*Ocimum gratissimum* leaf oil	Agar diffusion	[109]
	EO + Preservative	1	mint, oregano, rosemary, sage	Broth dilution	[110]
	Carboxymethyl cellulose film	2	*Zataria multiflora Boiss* oil	Agar diffusion	[96]
	Water-based emulsion	2	garlic oil	Agar diffusion	[100]
	EO + Emulsifier	1	oregano oil, cinnamon oil, tea tree oil, lavender oil	Agar diffusion	[111]
	Chitosan film	3	*Eucalyptus globulus* oil	Agar diffusion	[102]
	EO + Interfering substance	1	tea tree oil	Agar diffusion, broth dilution	[95]
	Gelatin film	2	oregano, lavender oil	Agar diffusion	[104]
	Cellulose film	2	CA, EUG	Vapor diffusion	[93]
*Serratia marcescens*	EO + Interfering substance	1	tea tree oil	Agar diffusion, broth dilution	[95]
*Salmonella* Typhimurium	Carboxymethyl cellulose film	2	*Zataria multiflora Boiss* oil	Agar diffusion	[99]
	Water-based emulsion	2	garlic oil	Agar diffusion	[100]
	EO + Interfering substance	1	tea tree oil	Agar diffusion, broth dilution	[95]
	Edible coating	2	ginger oil	Agar diffusion	[107]
*Salmonella* Enteritidis	Emulsion	1	thyme oil	Broth dilution	[105]
	Cellulose film	2	CA, EUG	Vapor diffusion	[93]

^1^ Major components are as follows: bonding agent, surfactant, thickener, emulsion stabilizer, emulsifier, ointment base, preservative, film former, plasticizer, detergent, cation, and organic substance. Solvents were excluded from the count. ^2^ Abbreviations of essential oils: cinnamaldehyde (CA), carvacrol (CAR), and eugenol (EUG).

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
