# Peer review of "Recent Advances in the Application of Antibacterial Complexes Using Essential Oils"

_molecules, 2020, doi:10.3390/molecules25071752_

Round 1

Reviewer 1 Report

This review manuscript covers recent studies of antibacterial activity of typical essential oils reported by many research groups. First of all, English proofreading for this article is strongly recommended. In Refs 43 and 81, the initial words of journal title should be upper case. Since the complementary effects of essential oils in the present results remains unclear, authors should find some way to explain the mechanism in molecular levels. Overall, I would recommend this review for publication in Molecules with minor revision.

Author Response

Thank you for the helpful comments to improve the quality of this manuscript.

Reviewer 2 Report

Line 18: long term

Line 37-38: EO are mainly secondary metabolites

Consider using the term "antibacterials" instead of antibiotics

Author Response

(The authors gave the same response as above.)

Reviewer 3 Report

The details revision ha been carried out diectly in the text find the attched pdf file

Author Response

(The authors gave the same response as above.)
